# SARS-CoV-2-specific T cells generated for adoptive immunotherapy are capable of recognizing multiple SARS-CoV-2 variants

**Archana Panikkar[1], Katie E. Lineburg[1], Jyothy Raju[1], Keng Yih Chew[2], George R. Ambalathingal[1], Sweera Rehan[1], Srividhya Swaminathan[1,3], Pauline Crooks[1], Laetitia Le Texier[1], Leone Beagley[1], Shannon Best[1], Matthew Solomon[1], Katherine K. Matthews[1], Sriganesh Srihari[1], Michelle A. Neller[1], Kirsty R. Short[2,4], Rajiv Khanna[1,3], Corey Smith[1,3]***

**1** QIMR Berghofer Centre for Immunotherapy and Vaccine Development and Translational and Human Immunology Laboratory, Department of Immunology, QIMR Berghofer Medical Research Institute, Herston, Queensland, Australia, **2** School of Chemistry and Molecular Biosciences, The University of Queensland, St Lucia Queensland, Australia, **3** Faculty of Medicine, The University of Queensland, Herston, Queensland, Australia, **4** Australian Infectious Diseases Research Centre, The University of Queensland, St Lucia Queensland, Australia

* corey.smith@qimrberghofer.edu.au

**Data Availability Statement:** All relevant data are within the manuscript and its Supporting Information files.

## Abstract

Adoptive T-cell immunotherapy has provided promising results in the treatment of viral complications in humans, particularly in the context of immunocompromised patients who have exhausted all other clinical options. The capacity to expand T cells from healthy immune individuals is providing a new approach to anti-viral immunotherapy, offering rapid off-the-shelf treatment with tailor-made human leukocyte antigen (HLA)-matched T cells. While most of this research has focused on the treatment of latent viral infections, emerging evidence that SARS-CoV-2-specific T cells play an important role in protection against COVID-19 suggests that the transfer of HLA-matched allogeneic off-the-shelf virus-specific T cells could provide a treatment option for patients with active COVID-19 or at risk of developing COVID-19. We initially screened 60 convalescent individuals and based on HLA typing and T-cell response profile, 12 individuals were selected for the development of a SARS-CoV-2-specific T-cell bank. We demonstrate that these T cells are specific for up to four SARS-CoV-2 antigens presented by a broad range of both HLA class I and class II alleles. These T cells show consistent functional and phenotypic properties, display cytotoxic potential against HLA-matched targets and can recognize HLA-matched cells infected with different SARS-CoV-2 variants. These observations demonstrate a robust approach for the production of SARS-CoV-2-specific T cells and provide the impetus for the development of a T-cell repository for clinical assessment.

## Author summary

Since the emergence of SARS-CoV-2 variants that reduce the effectiveness of vaccines, it is evident that other interventional strategies will be needed to treat COVID-19,

**Funding:** This work was supported by generous donations to the QIMR Berghofer COVID-19 appeal (CS, KEL, RK) and funding from the Queensland State Government (CS). SSw is supported by Australian Government Research Training Program Scholarship and RK is supported by a National Health and Medical Research Council Fellowship. The funders had no role in study design, data collection and analysis, decision to publish, or preparation of the manuscript.

**Competing interests:** The authors have declared that no competing interests exist.

particularly in patients with a compromised immune system who are at an increased risk of developing severe COVID-19. Off-the-shelf T-cell immunotherapy is proving to be a powerful tool to treat viral disease in patients with a compromised immune system. Here, we report here that a small number of SARS-CoV-2 exposed individuals can be used generate a bank of specific T cells that provide broad population coverage. Importantly, we demonstrate that most of the epitopes recognized by these T cells remain unchanged in different variants and that the T cells can recognize cells infected with three different variants of SARS-CoV-2. We believe these observations provide critical proof-of-concept that T-cell based immunotherapy may offer an option for the future treatment of immuno-compromised patients who remain susceptible to the severe complications associated with COVID-19.

## Introduction

T-cell immunotherapy is providing a paradigm shift in treatment options for patients with lethal infectious complications [1]. Research over the past three decades has demonstrated the great potential for the treatment of viral complications, particularly those that arise in immunocompromised individuals [2–4]. More recent studies have also shown the potential of T-cell immunotherapy in the treatment of non-viral infectious complications [5]. While early antiviral T-cell immunotherapy work was pioneered through the use of autologous or hematopoietic stem cell transplant donor peripheral blood mononuclear cells (PBMC) as a source of T cells, more recent approaches have transitioned to the use of cryopreserved T-cell repositories generated from PBMC from allogeneic healthy donors that have defined specificity to the pathogen of choice [6–8]. The advantage of this approach is the ability to rapidly administer T cells within days of diagnosis rather than the 2–4 weeks that is typically required to generate an autologous therapy. Donors can be specifically selected to provide desirable features, including CD4$^+$ and CD8$^+$ T cells capable of recognizing multiple viral antigens in order to minimize the risk of antigenic variant escape, and diverse human leukocyte antigen (HLA) coverage to allow the provision of T cells to a genetically diverse population.

The association of COVID-19 severity with age and other co-morbidities is a clear indication of greater risk to individuals with an increased level of immune compromise [9]. There is discordance between the efficient induction of T-cell immunity in patients who efficiently resolve COVID-19 and the T-cell dysfunction observed in patients with severe COVID-19. This discordance demonstrates the important role T cells likely play in disease control [10–13]. Whilst vaccination will be effective in a large percentage of the population [14,15], at-risk cohorts with compromised immunity (e.g. hematopoietic stem cell and organ transplant recipients) are less likely to generate effective immune responses. Multiple recent studies have demonstrated reduced vaccine-induced immunity in transplant recipients [16–20]. It is therefore likely that additional immunotherapeutic options will be needed for treatment. Based on our extensive experience with the administration of T cells to treat viral diseases in humans [3,21], and our recent characterization of critical immunological determinants in SARS-CoV-2 [22], we have developed a strategy to generate an allogeneic off-the-shelf T-cell repository from SARS-CoV-2-convalescent blood donors. Here we demonstrate that a small cohort of donors can be used to generate functionally potent SARS-CoV-2-specific T cells capable of recognizing multiple antigens and with broad HLA coverage across both HLA class I and class II alleles. Furthermore, we demonstrate that the T cell epitopes targeted in this approach are conserved and capable of recognizing cells infected with multiple variants of SARS-CoV-2.

## Results

### SARS-CoV-2-specific T cells can be readily generated from recovered individuals

To select potential donors for T cell immunotherapy, we assessed SARS-CoV-2 specific T cell responses from a cohort of sixty SARS-CoV-2-convalescent individuals recruited between 1 and 4 months after diagnosis, as described previously. The majority of these donors generated CD4$^+$ T cells responses against the spike (S), nucleocapsid (N) and membrane (M) proteins of SARS-CoV-2, while CD8$^+$ T-cell responses were directed against N, S and the ORF3a protein (Fig 1). Based on these analyses, we selected 12 participants who generated both CD4$^+$ and CD8$^+$ T-cell responses against multiple SARS-CoV-2 antigens, and who provided broad HLA coverage across common haplotypes in the population (Table 1). PBMC from these donors were stimulated with overlapping peptide pools from the four immunodominant antigens (N, S, M and ORF3a) and were then cultured for 2 weeks in the presence of IL-2 in G-Rex culture vessels. T-cell expansion and phenotype were monitored using a TBNK multi-test kit and cells were maintained at a density of $2 \times 10^6$ cells/cm$^2$. At the end of the expansion period, T cells were assessed for a series of functional attributes (Fig 1).

All cultures demonstrated cell expansion over the 14-day period with a median 18.4-fold expansion (range: 2.6–33.1 fold) (Fig 2A). Cultures were dominated by CD3$^+$CD4$^+$ T cells (median: 85.8%; range 68.6–93.4%), with the majority of the remaining cells being CD3$^+$CD8$^+$ T cells (median: 8.6%; range 2.6–27.9%) (Fig 2B). To assess the T-cell specificity, cultured cells were recalled with the overlapping SARS-CoV-2 peptide pools separately and assessed for

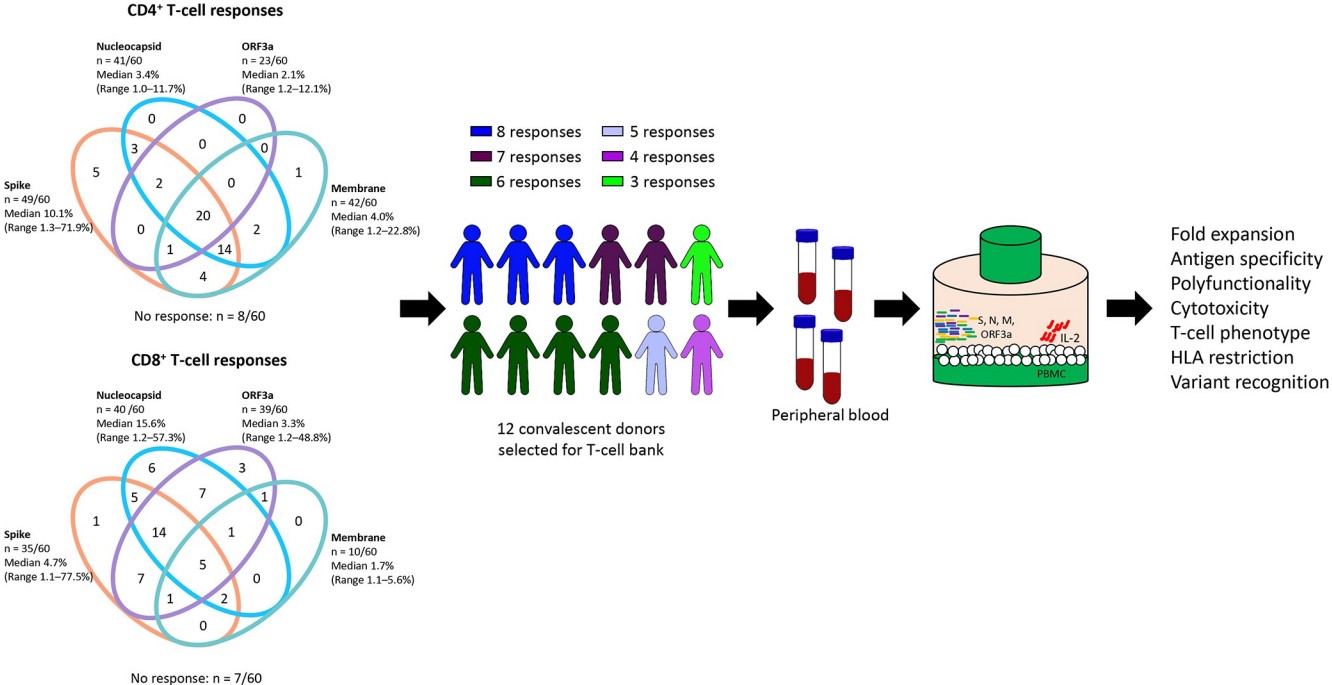

**Fig 1. Schematic representation of donor selection and generation of SARS-CoV-2-specific T cells for T-cell therapy.** Donors for T-cell expansion were selected from a cohort of 60 SARS-CoV-2-convalescent individuals whose cells had been screened for reactivity to SARS-CoV-2 antigens using an intracellular cytokine assay. Venn diagrams represent the number of donors generating a combination of antigen-specific responses by CD4$^+$ or CD8$^+$ T cells. Donors selected for T-cell expansion showed a minimum of three antigen-specific responses in CD4$^+$ and/or CD8$^+$ T cells. PBMC from each donor were stimulated with a single mixture of N, S, M and ORF3a overlapping peptide pools then cultured in G-Rex culture vessels for 2 weeks in the presence of IL-2. Cultured T cells were then assessed for a range of standard T-cell attributes.

**Table 1. HLA types of the SARS-CoV-2-convalescent participants included in the study.**

| Participant Code | Age | Sex | A | | B | | C | | DRB1 | | DPB1 | | DQB1 | | DQA1 | |
|---|---|---|---|---|---|---|---|---|---|---|---|---|---|---|---|---|
| Q-003 | 52 | M | 01:01 | 25:01 | 08:01 | 35:01 | 04:01 | 07:01 | 03:01 | 04:01 | 04:01 | | 02:01 | 03:01 | 03:03 | 05:01 |
| Q-004 | 30 | F | 02:01 | 03:01 | 07:02 | 44:02 | 05:01 | 07:02 | 07:01 | 15:01 | 04:01 | 04:02 | 02:02 | 06:02 | 01:02 | 02:01 |
| Q-005 | 33 | M | 02:01 | 11:01 | 15:01 | 35:01 | 03:04 | 04:01 | 01:01 | 04:01 | 04:01 | 04:02 | 03:02 | 05:01 | 01:01 | 03:01 |
| Q-006 | 48 | M | 02:01 | 24:02 | 07:02 | 18:01 | 07:01 | 07:02 | 04:01 | 07:01 | 02:01 | 20:01 | 02:02 | 03:01 | 02:01 | 03:03 |
| Q-012 | 71 | F | 02:01 | 03:01 | 44:02 | 51:01 | 01:02 | 05:01 | 01:01 | 04:01 | 03:01 | 04:02 | 03:01 | 05:01 | 01:01 | 03:03 |
| Q-014 | 61 | F | 03:01 | 29:02 | 07:02 | 44:03 | 07:02 | 16:01 | 07:01 | 15:01 | 04:01 | | 02:02 | 06:02 | 01:02 | 02:01 |
| Q-026 | 39 | F | 01:01 | 30:02 | 40:01 | 49:01 | 03:04 | 07:01 | 11:01 | 13:02 | 02:01 | 04:01 | 03:01 | 06:04 | 01:02 | 05:05 |
| Q-029 | 31 | F | 02:01 | 31:01 | 07:02 | 40:01 | 03:04 | 07:02 | 11:01 | 15:01 | 04:01 | 14:01 | 03:0 | 06:02 | 01:02 | 05:05 |
| Q-031 | 61 | M | 02:01 | 29:02 | 15:01 | 44:03 | 03:04 | 16:01 | 04:01 | 07:01 | 03:01 | 105:01 | 02:02 | 03:01 | 02:01 | 03:03 |
| Q-041 | 59 | M | 24:02 | 33:01 | 14:02 | 40:01 | 03:04 | 08:02 | 01:02 | 04:04 | 04:01 | 06:01 | 03:02 | 05:01 | 01:01 | 03:01 |
| Q-056 | 23 | M | 01:01 | | 37:01 | 57:01 | 06:02 | | 08:01 | 15:01 | 04:01 | | 04:02 | 06:02 | 01:02 | 04:01 |
| Q-058 | 52 | F | 02:05 | 11:01 | 35:01 | 49:01 | 04:01 | 07:01 | 01:02 | 01:03 | 02:01 | 04:02 | 05:01 | | 01:01 | |
| Q-008 | 49 | F | 02:01 | 03:01 | 07:02 | 44:02 | 07:02 | 07:04 | 04:04 | 15:01 | 01:01 | 04:01 | 03:02 | 06:02 | 01:02 | 03:01 |
| Q-024 | 57 | F | 01:01 | 03:01 | 07:02 | 08:01 | 07:01 | 07:02 | 03:01 | 15:01 | 03:01 | 04:01 | 02:01 | 06:02 | 01:02 | 05:01 |
| Q-002 (THI-COV-002) | 54 | F | 02:01 | 32:01 | 15:01 | 44:02 | 03:03 | 05:01 | 04:01 | 11:01 | 04:01 | 04:02 | 03:01 | 03:02 | 03:01 | 05:05 |
| Q-020 (THI-COV-003) | 59 | M | 24:02 | 31:01 | 07:02 | 35:02 | 04:01 | 07:02 | 11:04 | 15:01 | 04:01 | | 03:01 | 06:02 | 01:02 | 05:05 |

cytokine production using a multiparametric intracellular cytokine assay (S1 and S2 Figs). Due to the size of the S protein, two separate pools were used for recall (S-1 and S-2). All cultures exhibited a CD4$^+$ T-cell response (>1% above background) against the N, M and S proteins, whereas only five demonstrated a response against ORF3a (Fig 2C). While all cultures showed a CD8$^+$ T cell response (>1% above background) against at least one antigen, these responses were more focused than CD4$^+$ T-cell responses (Fig 2D). The CD8$^+$ T-cell responses were dominated by N-specific T cells (response in 11/12 cultures) and ORF3a-specific T cells (9/12 cultures). CD8$^+$ T cell responses to S (8/12 cultures) and M (5/12 cultures) were less frequent. To assess if responses to SARS-CoV-2 antigens could be generated from unexposed individuals, we selected ten unexposed volunteers and stimulated their PBMC with the overlapping peptide sets. No antigen-specific responses could be detected in this cohort (S3 Fig).

CD4$^+$ and CD8$^+$ T cells displayed a consistent polyfunctional profile, irrespective of the antigen specificity. CD4$^+$ T cells had consistent expression of TNF, with the majority also expressing IFN-γ, and a high proportion were triple positive for the expression of IFN-γ, TNF and IL-2 (Fig 2E). However, as is evident for most virus-specific T-cell populations, CD4$^+$ T cells displayed little evidence of degranulation (CD107a). Conversely, the majority of SARS-CoV-2-specific CD8$^+$ T cells degranulated and produced IFN-γ and TNF (Fig 2F). A median of 22% (range: 8.8–49.3%) of specific CD8$^+$ T cells produced all four cytokines. These observations are consistent with our recent findings in T cells generated for adoptive cellular therapy in other settings of human viral disease [3].

To demonstrate that SARS-CoV-2-specific T cells display cytotoxic potential against HLA-matched targets, we performed a real-time killing assay using the xCELLigence platform. Target cells, HEK293 (A*02:01, A*03:01, B*07:02, B*08:01, C*07:02, DRB1*15:01, DQB1*06:02) and F#62 (A*02:01, B*40:01, B*44, C*03, C*07), were adhered to xCELLigence plates overnight then pulsed with single-antigen overlapping peptide pools or left unpulsed. HLA-matched SARS-CoV-2-specific T cells were added to the target cells and killing was assessed over a 24 hour period. Two T-cell cultures were assessed, and both showed evidence of target-specific killing associated with antigen specificity of the culture. T cells from participant Q-029 killed HEK293 cells pulsed with N and S (Fig 2G), while participant Q-026's T cells induced lysis of

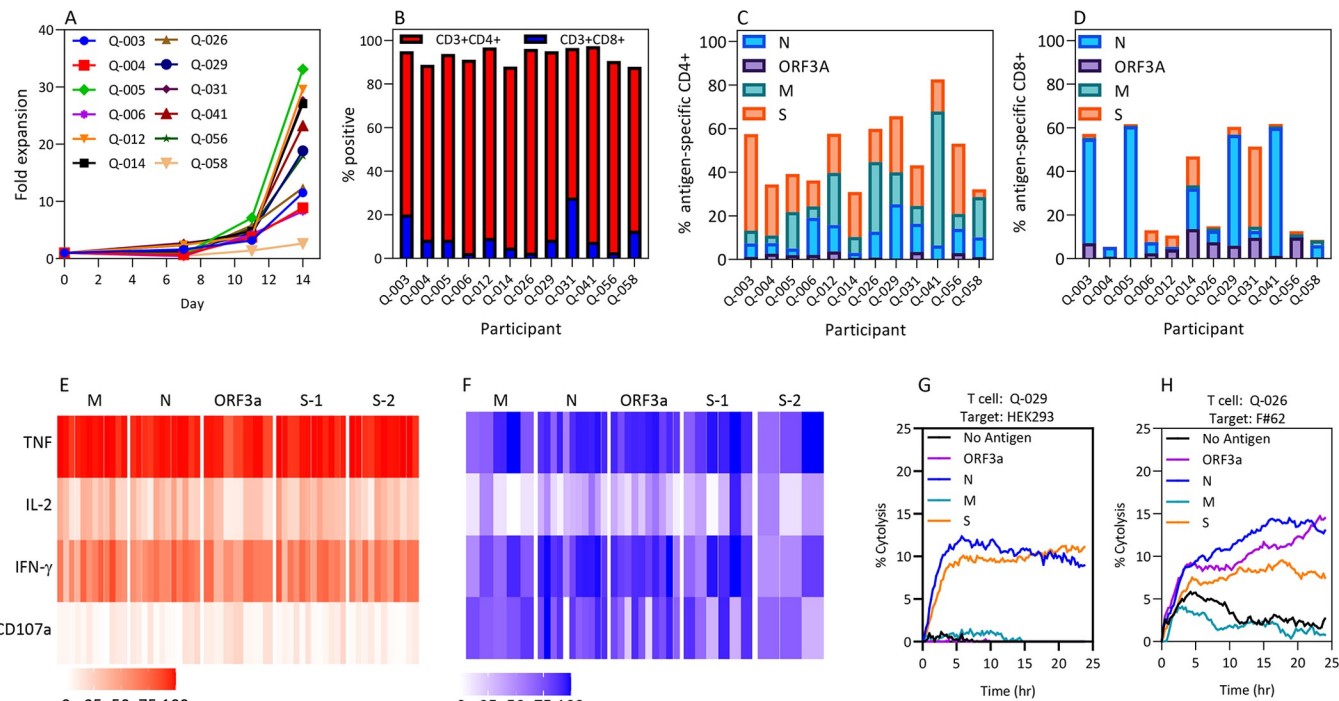

**Fig 2. Expansion of SARS-CoV-2-specific T cells from SARS-CoV-2-convalescent individuals.** PBMC from 12 individuals who had cleared SARS-CoV-2 infection were stimulated with overlapping peptide pools corresponding to the ORF3a, N, M and S antigens of SARS-CoV-2, then cultured for 2 weeks in the presence of IL-2 **(A)** CD45$^+$ cell yield was determined using Trucount tubes during the course of T-cell culture. Data represent fold expansion of T cells from each individual, relative to the total number of CD45$^+$ cells at the start of the culture period. **(B)** The phenotypic characteristics of the cultured T cells were assessed using standard TBNK analysis. Data represent the proportion of CD3$^+$CD4$^+$ and CD3$^+$CD8$^+$ T cells at the time of cell harvest. Cultured T cells were stimulated with the ORF3a, N, M and S (S-1 and S-2) overlapping peptide pools, then assessed for IFN-γ production using an intracellular cytokine assay. **(C)** Data represent the frequency of antigen-specific CD4$^+$ T cells in response to each overlapping peptide pool, following subtraction of the background response in the no-peptide control. **(D)** Data represent the frequency of antigen-specific CD8$^+$ T cells in response to each overlapping peptide pool, following subtraction of the background response in the no-peptide control. The response to the S protein was determined by combining the responses to the S-1 and S-2 pools. **(E)** Calendar plots represent the proportion of the total antigen-specific CD4$^+$ T cells from each participant that produced each effector function. **(F)** Calendar plots represent the proportion of the total antigen-specific CD8$^+$ T cells from each participant that produced each effector function. **(G)** HEK293 cells and **(H)** a human fibroblast cell line (F#62) were seeded in an E-plate overnight, then pulsed with the ORF3a, N, M and S peptide pools separately. T cells from Q-029, Q-003 and Q-041 were added to the corresponding HLA-matched targets at a 1:1 effector-to-target ratio. The proliferative index of the cells was acquired using the RTCA system for 72 hours. Cytolysis of the target cell lines with and without antigen was calculated based on the RTCA system specification. Data represent the mean deviation of cytolysis from three replicates for Q-029 T-cell lysis of HEK293 cells **(G)** and Q-026 T-cell lysis of F#62 fibroblasts.

the F#62 primary human fibroblast cell line pulsed with peptides from N, M and S antigens (Fig 2H). These observations demonstrate that SARS-CoV-2-specific T cells display functional characteristics typically associated with other in vitro-expanded antigen-specific T-cell therapy products.

To assess the profile of SARS-CoV-2-specific T cells, we employed an antibody panel designed to assess phenotypic (CD27, CD28, CD45RA and CD57), transcriptional (T-bet and Eomes) and functional (granzyme B and perforin) properties. The markers were selected to define T cells with an effector profile characterized by high expression of granzyme B, perforin, T-bet and Eomes, evidence of senescence using CD57, and presence of co-stimulatory markers CD27 and CD28. To define antigen-specific T cells, expanded T cells were stimulated with peptides from ORF3a, N, M or S antigen, then assessed for the expression of TNF (S4 Fig). TNF-expressing T cells from positive cultures were concatenated into a single file and t-SNE analysis employed to define distinct populations of T cells (Fig 3A). This analysis revealed four distinct populations, including an effector CD8$^+$ T-cell population (population 1) co-

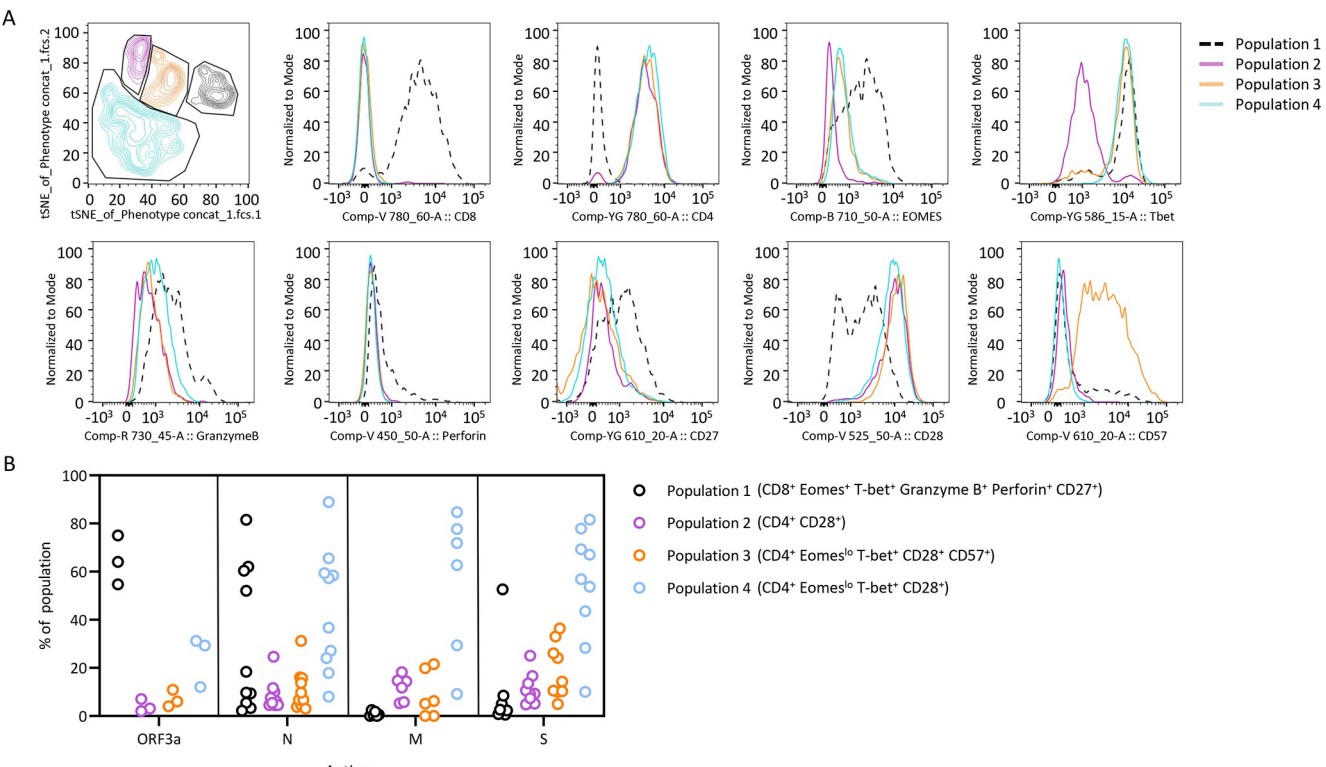

**Fig 3. Phenotypic profile of SARS-CoV-2-specific T cells.** SARS-CoV-2 peptide pool-stimulated T cells from responding donors (>1% IFN-γ$^+$) were assessed for expression of TNF. TNF-expressing T cells were given a sample number, concatenated into a single file and t-SNE analysis performed. **(A)** Dot plot represents the gating strategy used to define four distinct T-cell populations following t-SNE analysis. Histograms represent marker expression in each of the four populations. **(B)** Samples were separated based on sample ID and the proportion of antigen-specific T cells in each population was determined. Data represent the proportion of T cells in each population following stimulation with ORF3a, N, M and S.

expressing T-bet, Eomes, granzyme B, perforin and CD27, and three distinct populations of CD4$^+$ T cells that all expressed CD28, but displayed differential expression of Eomes, T-bet and CD57. Deconvolution of the samples revealed that while ORF3a- and N-specific T-cell populations contained both effector CD8$^+$ T cells and CD4$^+$ T cells, M- and S-specific T-cell populations were consistently dominated by CD4$^+$ Eomes$^{lo}$ T-bet$^+$ CD28$^+$ T cells (population 4). The M- and S-specific T cells also included two lower-frequency populations, which were likely to be poorly differentiated (population 2, CD4$^+$ CD28$^+$) and terminally differentiated (population 3, CD4$^+$ Eomes$^{lo}$ T-bet$^+$ CD28$^+$ CD57$^+$) CD4$^+$ T cells (Fig 3B).

## Broad HLA restriction of SARS-CoV-2-specific T cells

A critical aspect of any allogeneic T-cell bank is the ability to provide broad HLA coverage to allow administration to a diverse cross-section of the population. To define the HLA class I restriction of the SARS-CoV-2-specific T cells, we took advantage of a library of HLA-deficient K562 cells transfected with single HLA class I alleles (S1 Table). To define HLA class II restriction, we used a combination of HLA-deficient T2 cells transfected with HLA-DR alleles and Epstein–Barr virus-transformed lymphoblastoid cell lines (EBV-LCL) matched for single HLA-DR alleles with the T cells. Untransfected K562/T2 cells or HLA-mismatched EBV-LCL were used as controls. The target cells were pulsed with the overlapping peptide pools and used as antigen-presenting cells to HLA-matched SARS-CoV-2-specific T cells, and an intracellular IFN-γ assay was used to detect activation of CD4$^+$ or CD8$^+$ T cells. Representative

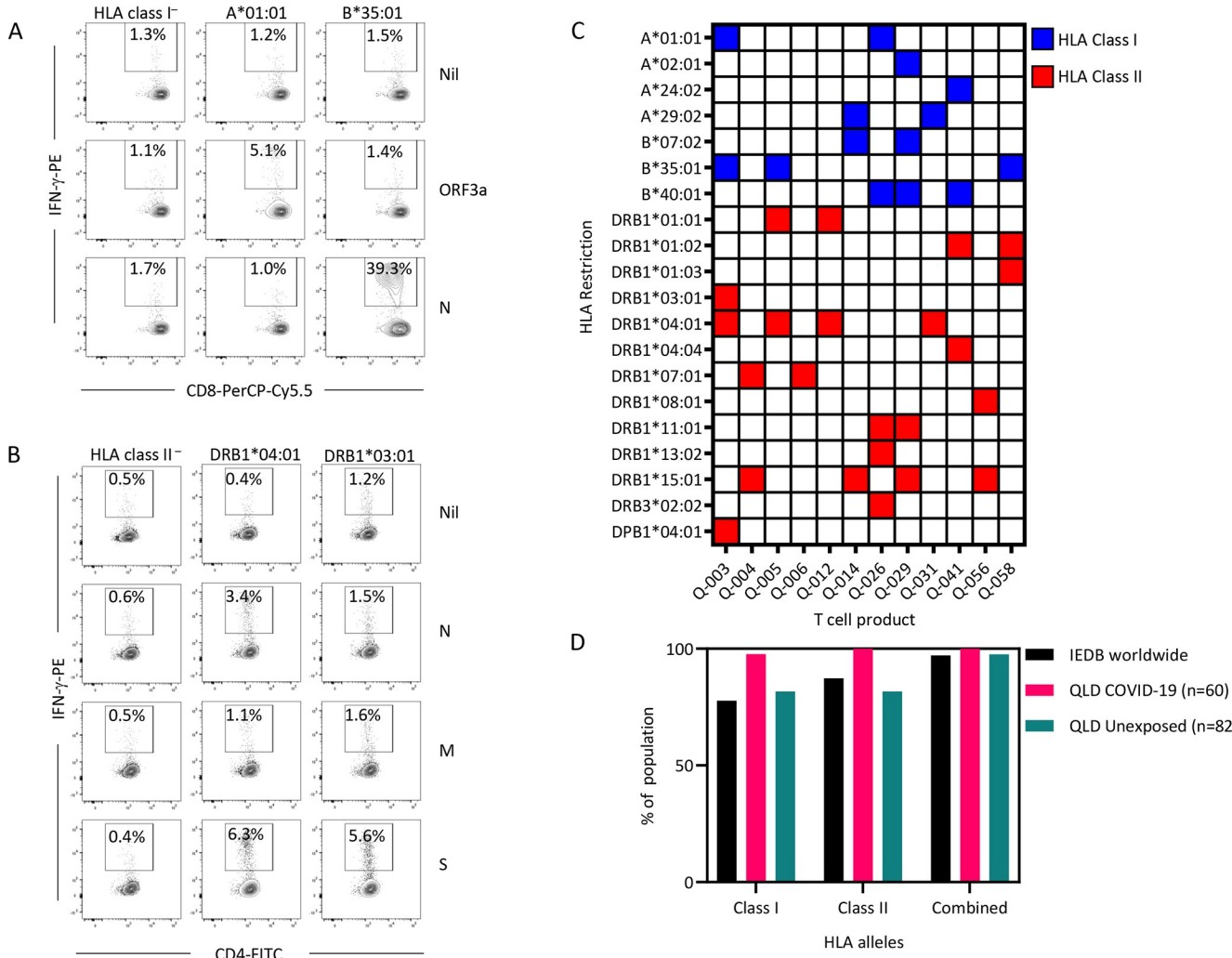

**Fig 4. HLA restriction of SARS-CoV-2-specific T cells.** SARS-CoV-2-specific T cells were exposed to HLA-matched and mismatched target cell lines (K562, PHA-blasts, T2 cells or EBV-LCL) pulsed with ORF3a, N, M and S overlapping peptide sets. T cells were assessed for activation by IFN-γ production. **(A)** Representative data from an HLA-restriction analysis of N- and ORF3a-specific CD8$^+$ T cells from a single donor (Q-003). Left panels represent T cells cultured with K562, middle panels represent T cells cultured with K562-A0101 and right panels T cell cultured with K562-B3501. **(B)** Representative HLA-restriction analysis of N-, M- and S-specific CD4$^+$ T cells from a single donor (Q-003). Left panels represent T cells cultured with T2, middle panels represent T cells cultured with T2-DR4 and right panels T cell cultured with T2-DR3. **(C)** Data represent the HLA restrictions of functional T cells detected in each culture. Responses were considered positive if >1% of CD4$^+$ T cells or CD8$^+$ T cells were IFN-γ$^+$ following subtraction of the control. Blue boxes represent positive HLA class I restrictions. Red boxes represent positive HLA class II restrictions. **(D)** Based upon the HLA restrictions detected in the HLA-restriction assays, HLA coverage was determined using the IEDB database (www.iedb.org), using our full cohort of convalescent individuals (n = 60) and a cohort of unexposed individuals recruited in Queensland, Australia (n = 82).

analysis of IFN-γ production from a single donor's T cells is shown in Fig 4. This donor (Q-003) demonstrated HLA class I-restricted T cells via HLA-A*01:01 (ORF3a) and HLA-B*35:01 (N), with no presentation by HLA class I-deficient cells (Fig 4A). HLA class II restriction of N-specific CD4$^+$ T cells was via HLA-DRB1*04:01, while the S-specific CD4$^+$ T cells were restricted via both HLA-DRB1*03:01 and HLA-DRB1*04:01 (Fig 4B). The M-specific response in this donor did not reach the 1% above background threshold. A summary of the HLA restrictions detected in all T cells is provided in Fig 4C and S2 Table. From the 12 donors, we were able to identify HLA class I-restricted T cells recognizing 7 alleles, and HLA class II-restricted T cells recognizing 13 alleles, including the highly conserved DRB3*02:02 allele.

HLA coverage analysis using the Immune Epitope Database (IEDB; www.iedb.org) revealed that these T cells could potentially provide 77.63% HLA class I coverage, 87.34% HLA class II coverage and a combined coverage of 97.15% worldwide (Fig 4D). Similarly, in a cohort of 60 SARS-CoV-2-convalescent individuals and 82 unexposed healthy individuals recruited in Australia, our T cells would potentially provide 100% and 97.6% coverage respectively. These observations demonstrate the potential of a small number of T-cell products to provide broad HLA coverage.

## Conservation of SARS-CoV-2 epitopes and T-cell recognition of cells infected with multiple variants of SARS-CoV-2

With the continual emergence of novel variants of concern, the efficacy of cellular therapy could be impacted by changes in epitope sequences. To compare epitope sequences, where possible we determined the immunodominant epitopes that corresponded with the HLA restrictions in each of the expanded T-cell populations (S5 Fig). These T-cell epitope sequences were then compared across the variants of concern (Table 2). We noted a high level of conservation in these determinants, including in the recently emerged Omicron variant. Of the 22 epitopes defined in our study, no evidence of mutation was evident in the Alpha, Beta, Gamma or Delta strains. This is consistent with previous studies on T-cell epitopes in SARS-CoV-2. We noted amino acid changes in three epitopes in the Omicron strain, suggesting the potential for escape from our T cells if patients are exposed to Omicron. While it is not yet clear if these changes render T cells less efficient, they suggest that a polyclonal approach targeting multiple antigens and epitopes will provide the most effective T-cell immunotherapy strategy against emerging variants.

**Table 2. Conservation of SARS-CoV-2 epitopes in variants of concerns.**

| D641G | Alpha | Beta | Gamma | Delta | Omicron |
|---|---|---|---|---|---|
| FTSDYYQLY (FTS) | FTSDYYQLY | FTSDYYQLY | FTSDYYQLY | FTSDYYQLY | FTSDYYQLY |
| YFTSDYYQLY (YFT) | YFTSDYYQLY | YFTSDYYQLY | YFTSDYYQLY | YFTSDYYQLY | YFTSDYYQLY |
| RIRGGDGKM (RIR) | RIRGGDGKM | RIRGGDGKM | RIRGGDGKM | RIRGGDGKM | RIRGGDGKM |
| SPRWYFYYL (SPR) | SPRWYFYYL | SPRWYFYYL | SPRWYFYYL | SPRWYFYYL | SPRWYFYYL |
| KPRQKRTAT (KPR) | KPRQKRTAT | KPRQKRTAT | KPRQKRTAT | KPRQKRTAT | KPRQKRTAT |
| MEVTPSGTWL (MEV) | MEVTPSGTWL | MEVTPSGTWL | MEVTPSGTWL | MEVTPSGTWL | MEVTPSGTWL |
| TPSGTWLTY (TPS) | TPSGTWLTY | TPSGTWLTY | TPSGTWLTY | TPSGTWLTY | TPSGTWLTY |
| YLQPRTFLL (YLQ) | YLQPRTFLL | YLQPRTFLL | YLQPRTFLL | YLQPRTFLL | YLQPRTFLL |
| YFPLQSYGF (YFP) | YFPLQSYGF | YFPLQSYGF | YFPLQSYGF | YFPLQSYGF | YFPLRSYSF |
| NFRVQPTESIVRFPN (NFR) | NFRVQPTESIVRFPN | NFRVQPTESIVRFPN | NFRVQPTESIVRFPN | NFRVQPTESIVRFPN | NFRVQPTESIVRFPN |
| GIIWVATEGALNTPK (GII) | GIIWVATEGALNTPK | GIIWVATEGALNTPK | GIIWVATEGALNTPK | GIIWVATEGALNTPK | GIIWVATEGALNTPK |
| NCTFEYVSQPFLMDL (NCT) | NCTFEYVSQPFLMDL | NCTFEYVSQPFLMDL | NCTFEYVSQPFLMDL | NCTFEYVSQPFLMDL | NCTFEYVSQPFLMDL |
| SKRSFIEDLLFNKVT (SKR) | SKRSFIEDLLFNKVT | SKRSFIEDLLFNKVT | SKRSFIEDLLFNKVT | SKRSFIEDLLFNKVT | SKRSFIEDLLFNKVT |
| LTDEMIAQYTSALLA (LTD) | LTDEMIAQYTSALLA | LTDEMIAQYTSALLA | LTDEMIAQYTSALLA | LTDEMIAQYTSALLA | LTDEMIAQYTSALLA |
| MIAQYTSALLAGTIT (MIA) | MIAQYTSALLAGTIT | MIAQYTSALLAGTIT | MIAQYTSALLAGTIT | MIAQYTSALLAGTIT | MIAQYTSALLAGTIT |
| CSNLLLQYGSFCTQL (CSN) | CSNLLLQYGSFCTQL | CSNLLLQYGSFCTQL | CSNLLLQYGSFCTQL | CSNLLLQYGSFCTQL | CSNLLLQYGSFCTQL |
| AALALLLLDRLNQLE (AAL) | AALALLLLDRLNQLE | AALALLLLDRLNQLE | AALALLLLDRLNQLE | AALALLLLDRLNQLE | AALALLLLDRLNQLE |
| LLQYGSFCTQLNRAL (LLQ) | LLQYGSFCTQLNRAL | LLQYGSFCTQLNRAL | LLQYGSFCTQLNRAL | LLQYGSFCTQLNRAL | LLQYGSFCTQLKRAL |
| GYYRRATRRIRGGDG (GYY) | GYYRRATRRIRGGDG | GYYRRATRRIRGGDG | GYYRRATRRIRGGDG | GYYRRATRRIRGGDG | GYYRRATRRIRGGDG |
| LRGHLRIAGHHLGRC (LRG) | LRGHLRIAGHHLGRC | LRGHLRIAGHHLGRC | LRGHLRIAGHHLGRC | LRGHLRIAGHHLGRC | LRGHLRIAGHHLGRC |
| TLVKQLSSNFGAISS (TLV) | TLVKQLSSNFGAISS | TLVKQLSSNFGAISS | TLVKQLSSNFGAISS | TLVKQLSSNFGAISS | TLVKQLSSKFGAISS |
| AVYRINWITGGIAIA (AVY) | AVYRINWITGGIAIA | AVYRINWITGGIAIA | AVYRINWITGGIAIA | AVYRINWITGGIATA | AVYRINWITGGIAIA |

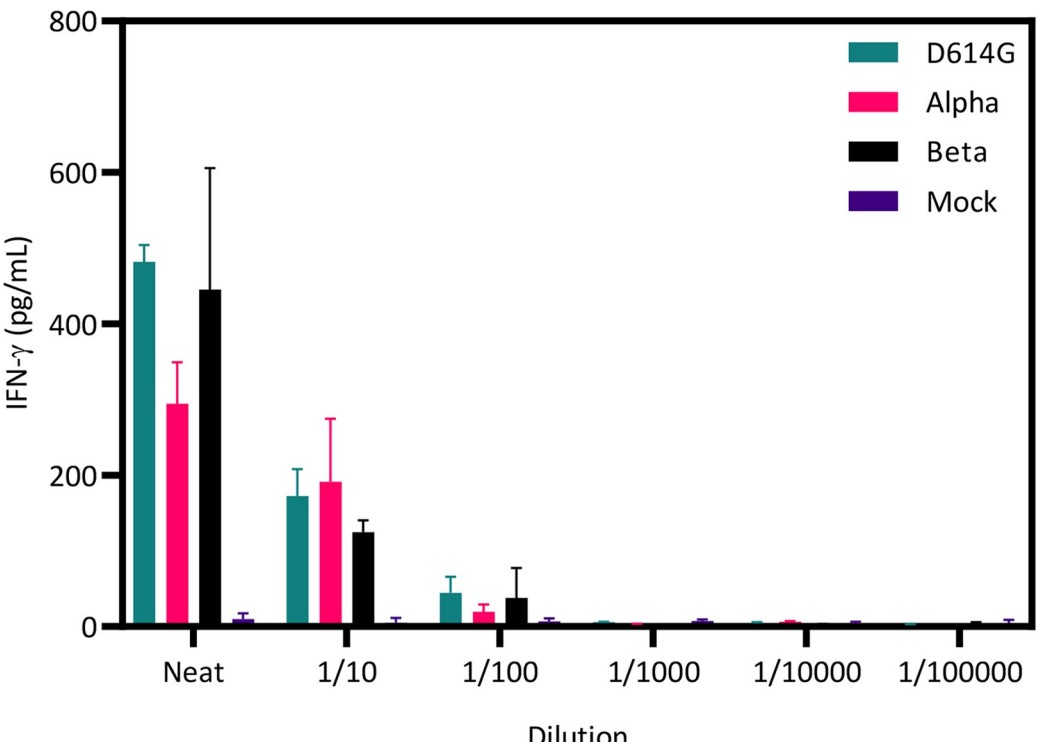

**Fig 5. Recognition of multiple SARS-CoV-2 variants by T cells generated for adoptive therapy.** HEK293T cells expressing ACE2 and TMPRSS2 were infected in duplicate with serial dilutions of the D614G, Alpha and Beta SARS-CoV-2 variants. Two hours after infection, T cells generated from donor Q-029 were added to duplicate wells containing infected and mock-infected HEK293 cells at a ratio of 10:1. Cells were incubated for 48 hours and then supernatants were assessed for IFN-γ using a standard ELISA. IFN-γ concentration was interpolated from a standard curve using sigmoidal nonlinear regression in GraphPad Prism.

To determine if in vitro-expanded T cells could recognize different variants of SARS-CoV-2, we infected HEK293-ACE2/TMPRSS2 cells (A\*02:01, B\*07:02, DRB1\*15:01) with serial dilutions of the D614G, Alpha and Beta variants. Two hours after infection, HLA-matched SARS-CoV-2-specific T cells from Q-029 (which included A\*02:01-, B\*07:02- and DRB1\*15:01-restricted virus-specific T cells) were added to the infected and mock-infected cells and incubated for 48 hours. Supernatant was harvested and assessed for IFN-γ production by ELISA. We detected IFN-γ secretion following incubation of the Q-029 T cells with all three viral variants (Fig 5), while no IFN-γ was detected following co-culture with the mock-infected cells. These observations are consistent with our in vitro killing assessment of Q-029 T cells, using peptide-pulsed HEK293 cells as targets, and suggest that the recognition of conserved epitopes by SARS-CoV-2-specific T cells has the potential to provide a second line of defense against viral escape from the humoral immune response.

## Generation of clinical-grade SARS-CoV-2-specific T cells

Our preclinical observations demonstrated that we had developed a robust approach to generate SARS-CoV-2-specific T cells across a broad range of HLA types. To refine our approach for clinical manufacture, we custom-designed two peptide pools based upon T-cell epitopes defined from our larger cohort of 60 volunteers. The first pool contains 32 peptides of 8–11 amino acids in length that encode a panel of defined CD8+ T cell epitopes from SARS-CoV-2 antigens (S3 Table), while the second contains 52 peptides of 15 amino acids in length that

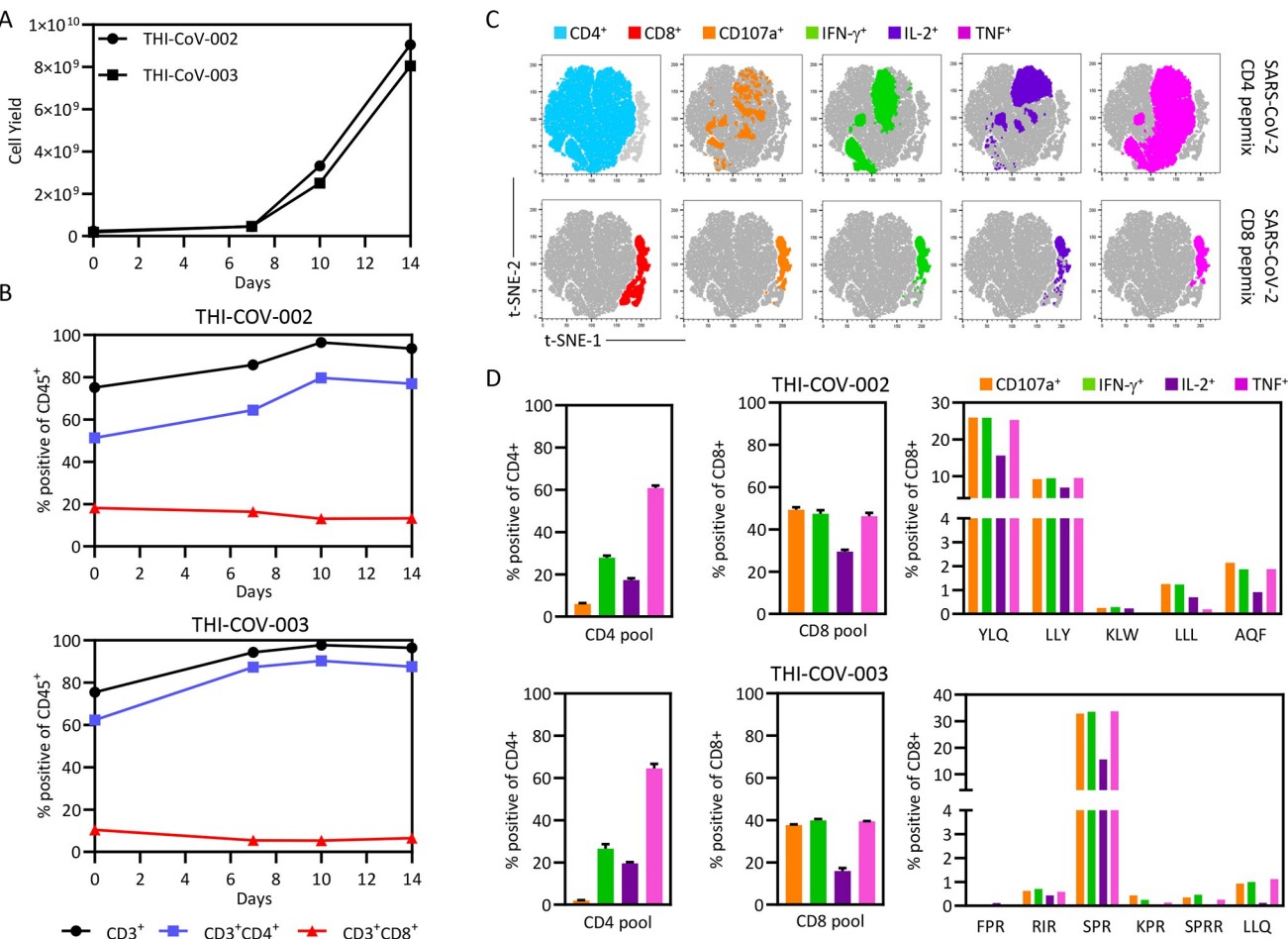

**Fig 6. Clinical-grade manufacture of SARS-CoV-2-specific T cells.** Two SARS-CoV-2-convalescent volunteers donated 400 mL of blood via venesection. PBMC were isolated from the blood and cryopreserved. Cryopreserved PBMC were thawed and stimulated with the custom-designed SARS-CoV-2 CD4 and CD8 peptide pools, then cultured for 14 days in the presence of IL-2 to generate T cell products THI-COV-002 and THI-COV-003. On day 14, cells were cryopreserved for future clinical use. Cultured cells were counted and phenotyped on days 7, 11 and 14 using a standard TBNK assessment. **(A)** Data represent the number of viable CD45[+] cells at each time point. **(B)** Data represent the proportion of CD3[+], CD3[+]CD8[+] and CD3[+]CD4[+] T cells at each time point. On the day of harvest, a polyfunctional intracellular cytokine assay was performed to assess specificity towards the CD4 and CD8 peptide pools and HLA class I-matched CD8[+] T cell epitopes. **(C)** Data represent t-SNE analysis of concatenated CD4[+] and CD8[+] T cells from triplicate samples stimulated with CD4 or CD8 peptide pools, overlaid with the expression of CD107a, IFN-γ, TNF or IL-2. **(D)** Data represent the proportion of CD4[+] T cells producing cytokines in response to the CD4 peptide pool or CD8[+] T cells producing cytokines in response to the CD8 peptide pool or individual HLA-matched peptide epitopes. Error bars represent standard deviation from triplicate cultures.

encode CD4[+] T cell epitopes (S4 Table). Following validation that the peptide pools could expand both SARS-CoV-2-specific CD4[+] and CD8[+] T cells in the laboratory from convalescent donor blood samples collected 12 months after infection (S6 Fig), we generated two clinical-grade lots of SARS-CoV-2-specific T cells (THI-COV-002 and THI-COV-003) under Good Manufacturing Practice conditions at Q-Gen Cell Therapeutics. T cells were expanded following stimulation with the SARS-CoV-2 CD4[+] and CD8[+] peptide pools, and assessed for cell yield, phenotype and viability using standard TBNK analysis every 3–4 days. T cells were assessed for antigen specificity and cryopreserved (110 vials at $4 \times 10^7$ cells/vial) after 14 days in culture. Both cell products displayed similar cell expansion kinetics (Fig 6A), with a mean 40-fold expansion after 14 days. Similarly to our earlier analyses (Fig 2B), THI-COV-002 and THI-COV-003 were dominated by CD4[+] T cells, with lower frequencies of CD8[+] T cells in each product (Fig 6B). While SARS-CoV-2-specific CD4[+] and CD8[+] T cells are capable of

generating multiple cytokines (Fig 6C), SARS-CoV-2-specific CD4+ T cells in both products (Fig 6D and 6E) displayed preferential production of TNF and limited degranulation, whereas CD8+ T cells displayed a consistent polyfunctional profile characterized by equivalent production of TNF, IFN-γ and CD107a. CD8+ T cells in THI-COV-002 preferentially recognized the HLA-A*02:01-restricted YLQ (S-encoded) and LLY (ORF3A-encoded) epitopes, while CD8+ T cells in THI-COV-003 were dominated by T cells recognizing the HLA-B*07:02-restricted SPR (N-encoded) epitope. These observations demonstrate the feasible translation of SARS-CoV-2-specific T cell expansion for clinical manufacture.

## Discussion

Emerging evidence suggests that T cells play an important role in the prevention of COVID-19 [10,12,13]. Adoptive cellular immunotherapy with SARS-CoV-2-specific T cells may provide an effective mechanism to control COVID-19. In this study, we have demonstrated a robust approach to the generation of SARS-CoV-2-specific T cells from convalescent individuals. Importantly, we have shown that these T cells display consistent functional properties that are typical of human virus-specific T cells [3,23,24]. We also extend on recent observations [25,26], demonstrating that a small cohort of volunteers can provide greater than 95% worldwide population coverage and that these T cells are capable of recognizing cells infected with multiple viral variants. These observations provide optimism that SARS-CoV-2-specific T cell products will be therapeutically beneficial, irrespective of the infecting strain.

Ongoing vaccination programs will have a profound impact upon the incidence of COVID-19 in the next few years [14,15]. However, it is not clear if vaccination will prevent the continued worldwide transmission of SARS-CoV-2, and it is likely that some individuals, particularly immunocompromised patients and the elderly, will remain susceptible to COVID-19 [27,28]. Considering the ineffectiveness of current anti-viral therapies and the potential for immune escape from antibody-based immunotherapies due to the development of novel mutations [29,30], other therapeutic options are needed. T cells that target more conserved regions of SARS-CoV-2 offer some advantages over sera and monoclonal antibodies and are currently being investigated by a number of groups as a potential therapeutic option for COVID-19 [26].

Virus-specific T-cell immunotherapies, which are currently in phase II and III studies, have emerged as an effective means to treat virus-associated complications, particularly against persistent viruses in transplant patients with compromised immunity [2,4]. While there is less evidence for their efficacy in humans against respiratory viruses, for which clinical trials have been limited to adenoviral infections [31], studies in mice have shown that adoptive therapy with CD8+ and/or CD4+ T cells can efficiently control respiratory infections, including in models of SARS-CoV-1 [32–34]. One limitation associated with the broad application of anti-viral T-cell immunotherapy is the time required to generate autologous T cells, which would likely preclude their use in settings of acute respiratory infections that induce disease rapidly. However, recent success in the application of HLA-matched, off-the-shelf, banked T cells for the treatment of a number of viral complications suggests that a similar strategy could be used in the setting of COVID-19 [8,31,35]. Our observations that T cells can be generated from a small cohort of convalescent individuals to cover a large percentage of the population suggest that the manufacture of a similar repository of SARS-CoV-2-specific T cells is feasible. Other groups have also developed approaches for a T-cell immunotherapy approach to COVID-19 [36–38], with the first case report of a heart transplant patient who received SARS-CoV-2 specific T cell therapy recently published [39].

This study shows clear evidence of immunodominant hierarchies in SARS-CoV-2-specific CD8+ T cells. The immunodominant hierarchies have the potential to impact the therapeutic

benefit in a portion of future recipients. While we saw immunodominance of T cells restricted via HLA-B*07:02, HLA-B*35:01, HLA-B*40:01 and HLA-A*29:01, we noted a generally poor anti-SARS-CoV-2 response via HLA-A*02:01. This is consistent with recent reports on HLA-A*02:01-restricted T cell responses in COVID-19 patients [40]. Similarly, we saw only limited reactivity through HLA-A*24:02, a common allele worldwide, and did not identify responses via other common HLA class I alleles, including HLA-A*03:01 and HLA-A*11:01. These observations are likely a consequence of bias in the ethnicity and HLA types of our donor cohort and suggest that broadening the HLA class I coverage of our T-cell bank will require assessment of T-cell responses in a more HLA class I-diverse cohort of convalescent individuals. In contrast, we saw less evidence of this bias in the CD4$^+$ T-cell response, and were able to demonstrate broader HLA class II restriction, particularly against alleles that are dominant worldwide.

The role that adoptive T-cell immunotherapy will play as a treatment option for COVID-19 remains to be determined. However, our observations demonstrate a robust approach for the generation of HLA-diverse SARS-CoV-2-specific T cells from convalescent individuals, providing the impetus for the production of clinical-grade material and further assessment in clinical trials.

## Materials and methods

### Ethics statement

This study was performed according to the principles of the Declaration of Helsinki. Ethics approval to undertake the research was obtained from the QIMR Berghofer Medical Research Institute Human Research Ethics Committee. Written informed consent was obtained from all participants.

### Study participants

The inclusion criteria for the study were that participants were over the age of 18, had been clinically diagnosed by PCR with SARS-CoV-2 infection, and had subsequently been released from isolation following resolution of symptomatic infection. A total of 60 participants were assessed in this study. Blood samples were collected from all participants and peripheral blood mononuclear cells (PBMC) were isolated and cryopreserved. HLA typing was performed on each donated sample. The HLA types of study participants are provided in Table 1.

### Expansion of SARS-CoV-2-specific T cells

Cryopreserved PBMC were rapidly thawed, then one-third incubated with peptide pools containing 15-mer peptides with 11 amino acid overlap corresponding to ORF3a (PM-WCPV-APA3-1), nucleocapsid (N; PM-WCPV-NCAP-1), membrane (M, PM-WCPV-VME-1) and spike (S, PM-WCPV-S-1) protein sequences from SARS-CoV-2 (JPT Technologies). For clinical-grade manufacture, two custom peptide pools containing defined CD8$^+$ and CD4$^+$ T cell epitopes (Mimotopes) were used. PBMC were washed, then mixed with the remaining cognate PBMC and seeded at a density of between 2 and $3 \times 10^6$ cells/cm$^2$ in G-Rex 6-well culture plates (Wilson Wolf). Cells were grown in RPMI supplemented with 5% human AB serum at 37˚C/6.5% CO$_2$/95% relative humidity. Recombinant IL-2 (120 IU/mL) was added on day 3 and supplemented every 2–3 days thereafter. On days 7 and 11, cultured cells were phenotyped and enumerated using the Multitest 6-Color TBNK Reagent (BD Biosciences), and divided into wells at a density between 2 and $3 \times 10^6$ cells/cm$^2$. On day 14, T cells were harvested, phenotyped and enumerated using the Multitest 6-Color

TBNK Reagent and assessed for antigen specificity using intracellular cytokine analysis as outlined below. The remaining cells were cryopreserved in culture media with 10% dimethyl sulfoxide (WAK-Chemie Medical). Clinical-grade manufacture was performed under GMP conditions at Q-Gen Cell Therapeutics.

## Intracellular cytokine assay

Cultured T cells were stimulated separately with the SARS-CoV-2 overlapping peptide pools and incubated for 4 hours at 37˚C in the presence of GolgiPlug, GolgiStop and anti-CD107a-FITC (BD Biosciences). Following stimulation, cells were washed and stained with anti-CD8-PerCP-Cy5.5 (eBioscience), anti-CD4-Pacific Blue (BD Biosciences) and live/dead fixable near-IR dead cell stain (Life Technologies) for 30 minutes at 4˚C before being fixed and permeabilized with Fixation/Permeabilization solution (BD Biosciences). After 20 minutes of fixation, cells were washed in BD Perm/Wash buffer (BD Biosciences) and stained with anti-IFN-γ-Alexa Fluor 700, anti-IL-2-PE and anti-TNF-APC (all from BD Biosciences) for a further 30 minutes at 4˚C. Finally, cells were washed again and acquired using a BD LSRFortessa with FACSDiva software. Post-acquisition analysis, including t-distributed stochastic neighbor embedding (t-SNE), was performed using FlowJo software (TreeStar). Cytokine detection levels identified in the no-peptide control condition were subtracted from the corresponding test conditions to account for non-specific, spontaneous cytokine production. T cells were deemed reactive to an antigen if >1% of $CD4^+$ or $CD8^+$ T cells produced IFN-γ after subtraction of background.

## Phenotypic analysis of T cells

Cultured T cells were stimulated separately with the SARS-CoV-2 overlapping peptide pools, corresponding to ORF3a, N, M and S proteins from SARS-CoV-2, and incubated at 37˚C for 4 hours in the presence of GolgiPlug. Following stimulation, cells were washed and stained with anti-CD8-BV786, anti-CD4-PE-Cy7, anti-CD27-PE/Dazzle 594, anti-CD28-BV480 (BD Biosciences), anti-CD57-BV605, anti-CD45RA-FITC (BD Biosciences) and live/dead fixable near-IR dead cell stain (Life Technologies) for 30 minutes at 4˚C before being fixed and permeabilized with TF Fix/Perm buffer (BD Pharmingen). After 1 hour of fixation, cells were washed in TF Perm/Wash buffer (BD Pharmingen) and stained with anti-TNF-APC (BD Biosciences), anti-Tbet-PE (eBioscience), anti-Eomes-PerCP-eFluor 710 (eBioscience), anti-Granzyme B-AF700 (BD Biosciences) and anti-Perforin-BV421 (Biolegend) for a further 30 minutes. Finally, cells were washed again and acquired using a BD LSRFortessa with FACSDiva software. Post-acquisition analysis was performed using FlowJo software (TreeStar). Gating analysis for the detection of antigen-specific TNF-producing cells is outlined in S1 Fig. TNF-producing cells from positive assays were then concatenated and t-SNE analysis was performed using the following parameters (Eomes, T-bet, CD27, CD28, CD57, granzyme B and perforin).

## HLA-restriction assay

Cryopreserved T cells were thawed in RPMI containing 10% fetal bovine serum and recombinant IL-2 (120 IU/mL), and incubated overnight at 37˚C prior to use. Target cell lines, including single HLA-transfected K562 and T2 cells, and EBV-LCL (S1 Table), were thawed and incubated overnight in RPMI containing 10% fetal bovine serum. Target cells were incubated with individual SARS-CoV-2 overlapping peptide pools for 1 hour at 37˚C, then washed to remove excess peptide. T cells were incubated with either HLA-matched peptide-pulsed or unpulsed target cells at a ratio of 10:1 for 4 hours in the presence of GolgiPlug. Peptide-pulsed and unpulsed HLA-deficient K562 or T2 cells, or HLA-mismatched EBV-LCL, were used as

controls. Following stimulation, cells were washed and stained with anti-CD8-PerCP-Cy5.5 (eBioscience), anti-CD3-APC anti-CD4-FITC and live/dead fixable near-IR dead cell stain for 30 minutes at 4˚C before being fixed and permeabilized with BD Fixation/Permeabilization solution (BD Biosciences). After 20 minutes of fixation, cells were washed in BD Perm/Wash buffer (BD Biosciences) and stained with anti-IFN-γ-PE for a further 30 minutes. Cells were washed again and acquired using a BD LSRFortessa with FACSDiva software. Post-acquisition analysis was performed using FlowJo software (TreeStar). Cytokine detection levels identified in the no-peptide control condition were subtracted from the corresponding test conditions to account for non-specific, spontaneous cytokine production. T cells were deemed restricted via an HLA molecule if >1% of CD4$^+$ or CD8$^+$ T cells produced IFN-γ after subtraction of background.

## Real-time T-cell killing assay

The xCELLigence RTCA MP Instrument (ACEA Biosciences) method was used to measure the cytotoxic potential of SARS-CoV-2-specific cells. Targets were seeded at a density of $5 \times 10^4$ (HEK293) or $1 \times 10^4$ (IFN-γ-activated primary fibroblast F#62) cells per well of the E-plate in RPMI containing 10% fetal bovine serum. The plates were kept in the incubator for 30 minutes and then transferred to the instrument inside a 37˚C incubator. Data were collected at 30-minute intervals for the entirety of the assay. Following overnight incubation, target cells were pulsed with 1 μg/mL of one individual SARS-CoV-2 overlapping peptide antigen, then incubated for 1 hour at 37˚C. Cells were washed to remove excess peptide, followed by the addition of T cells at an effector:target ratio of 5:1. The E-plates were kept in the laminar flow cabinet for 30 minutes before being transferred to the RTCA instrument. Data acquisition was resumed, and the cell index value was measured for 24 hours.

## Recognition of HEK293-ACE2 cells infected with multiple isolates of SARS-CoV-2

HEK-293T that stably express ACE2 and TMPRSS2 were kindly provided by Dr Giuseppe Baliatreri [41]. Cells were infected with serial dilutions of a D614G variant, hCoV-19/Australia/QLDID935/2020 (QLD935; GISAID accession *EPI_ISL_436097*), an alpha variant, hCoV-19/Australia/ QLD1517/2020 (QLS1517; GISAID accession EPI_ISL_944644), or a beta variant, hCoV-19/Australia/QLD1520/2020 (QLD1520; GISAID accession EPI_ISL_968081), or were mock infected. SARS-CoV-2 isolates were kindly provided by Queensland Health Forensic and Scientific Services. All work with infectious SARS-CoV-2 was performed under BSL3 conditions. Two hours later, in vitro-expanded SARS-CoV-2-specific T cells were added at a responder to stimulator ratio of 10:1. Cells were co-incubated for 48 hours, then supernatant harvested. The concentration of released IFN-γ was determined using the IFN-γ (human) AlphaLISA Detection Kit (PerkinElmer, Waltham, MA) as per manufacturer's instructions. IFN-γ concentration was interpolated from a standard curve using sigmoidal nonlinear regression in GraphPad Prism. 8.2.1 (San Diego, CA).

## Statistical analysis

GraphPad Prism 8.2.1 (San Diego, CA) was used to perform statistical analysis.

## Supporting information

**S1 Fig. Representative analysis of cytokine production by SARS-CoV-2-specific CD4$^+$ T cells.**
(TIF)

**S2 Fig. Representative analysis of cytokine production by SARS-CoV-2 specific CD8[+] T cells.**
(TIF)

**S3 Fig. Expansion of SARS-CoV-2 specific T cells in SARS-CoV-2-unexposed individuals.**
(TIF)

**S4 Fig. Gating strategy used to identify SARS-CoV-2-specific T cells by TNF expression.**
(TIF)

**S5 Fig. Summary of peptide epitope responses in T cells from convalescent participants.**
The three-letter code and antigen for each HLA-restricted epitope is provided, where defined.
(TIF)

**S6 Fig. Laboratory assessment of SARS-CoV-2 CD4 and CD8 peptide pools.** (A) Panels represent ICS analysis of expanded T cells from three donors recalled with and without the custom-designed SARS-CoV-2 peptide pools. (B) Comparison of SARS-CoV-2 specificity following expansion with either the overlapping peptide pools (OPP) or defined CD4 and CD8 peptide pools (Epitopes).
(TIF)

**S1 Table. Target cells used in the HLA-restriction assay.**
(DOCX)

**S2 Table. Summary of HLA-restriction analysis.**
(DOCX)

**S3 Table. SARS-CoV-2 CD8 Peptide Pool.**
(DOCX)

**S4 Table. SARS-CoV-2 CD4 Peptide Pool.**
(DOCX)

## Acknowledgments

We would like to thank all of the participants who generously donated their blood for this study.

## Author Contributions

**Conceptualization:** Archana Panikkar, Katie E. Lineburg, Jyothy Raju, George R. Ambalathingal, Katherine K. Matthews, Michelle A. Neller, Kirsty R. Short, Rajiv Khanna, Corey Smith.

**Data curation:** Archana Panikkar, Keng Yih Chew, Sriganesh Srihari, Kirsty R. Short, Corey Smith.

**Formal analysis:** Archana Panikkar, Sriganesh Srihari, Corey Smith.

**Funding acquisition:** Katie E. Lineburg, Rajiv Khanna, Corey Smith.

**Investigation:** Archana Panikkar, Katie E. Lineburg, Jyothy Raju, Keng Yih Chew, George R. Ambalathingal, Sweera Rehan, Srividhya Swaminathan, Pauline Crooks, Laetitia Le Texier, Leone Beagley, Shannon Best, Matthew Solomon, Corey Smith.

**Methodology:** Archana Panikkar, Jyothy Raju, Keng Yih Chew, George R. Ambalathingal, Corey Smith.

**Project administration:** Katherine K. Matthews, Michelle A. Neller, Corey Smith.

**Resources:** Rajiv Khanna.

**Supervision:** Michelle A. Neller, Kirsty R. Short, Rajiv Khanna, Corey Smith.

**Writing – original draft:** Archana Panikkar, Corey Smith.

**Writing – review & editing:** Archana Panikkar, Katie E. Lineburg, Jyothy Raju, Keng Yih Chew, George R. Ambalathingal, Sweera Rehan, Srividhya Swaminathan, Pauline Crooks, Laetitia Le Texier, Leone Beagley, Shannon Best, Matthew Solomon, Katherine K. Matthews, Sriganesh Srihari, Michelle A. Neller, Kirsty R. Short, Rajiv Khanna.

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
