## [Decision Letter · Decision Letter 0]

27 Jan 2022

Dear Dr. Smith,

Thank you very much for submitting your manuscript "SARS-CoV-2-specific T cells generated for adoptive immunotherapy are capable of recognizing multiple SARS-CoV-2 variants" for consideration at PLOS Pathogens. As with all papers reviewed by the journal, your manuscript was reviewed by members of the editorial board and by several independent reviewers. The reviewers appreciated the attention to an important topic. Based on the reviews, we are likely to accept this manuscript for publication, providing that you modify the manuscript according to the review recommendations.

The authors addressed the previous reviewers comments and the manuscript is substantially improved. Reviewer 1 has concerns about lack of novelty, but given the ongoing COVID19 pandemic, this work contributes to the wider field of understanding T cell responses to SARS-CoV-2. Please address the comments from Reviewer 1 regarding addition of latest references.

Sincerely,

Amy L Hartman, PhD

Associate Editor

PLOS Pathogens

Shin-Ru Shih

Section Editor

PLOS Pathogens

Kasturi Haldar

Editor-in-Chief

PLOS Pathogens

orcid.org/0000-0001-5065-158X

Michael Malim

Editor-in-Chief

PLOS Pathogens

orcid.org/0000-0002-7699-2064

The authors addressed the previous reviewers comments and the manuscript is substantially improved. Reviewer 1 has concerns about lack of novelty, but given the ongoing COVID19 pandemic, this work contributes to the wider field of understanding T cell responses to SARS-CoV-2. Please address the comments from Reviewer 1 regarding addition of latest references.

Reviewer Comments (if any, and for reference):

Reviewer's Responses to Questions

**Part I - Summary**

Reviewer #1: The authors present a revised version that is substantially improved from the original submission describing an off-the=-shelf heterologous T cell therapeutic for COVID-19. It is still my opinion that the notion of using heterologous T cell therapy as a viable means to treat COVID-19 is a bit limited in actual practicallity and potential utility. Nevertheless, the authors make a good case for potential, and adequately addressed the issues previously raised by the reviewers. Unfortunately, the study still lacks novelty, and the idea of using heterologous T cells for SARS-CoV-2 is not far off of using this approach for other viruses, and it is a mere incremental advancement to target this current hot viral target. Moreover, others have unfortunatly already published similar papers covering the same idea, diminishing the value of this new report.

**Part II – Major Issues: Key Experiments Required for Acceptance**

Reviewer #1: There are a number of papers that have come out in the past year, some of them more recently, that probably should now be acknowledged in the text and and properly cited in the manuscript. I've listed the PMID numbers: PMID: 33718360; PMID: 33905481; PMID: 34910857, PMID: 35003063

**Part III – Minor Issues: Editorial and Data Presentation Modifications**

Reviewer #1: (No Response)

PLOS authors have the option to publish the peer review history of their article (what does this mean?). If published, this will include your full peer review and any attached files.

Reviewer #1: No

Figure Files:

Data Requirements:

Reproducibility:

References:

---

## [Editor Report · Decision Letter 1]

4 Feb 2022

Dear Dr. Smith,

We are pleased to inform you that your manuscript 'SARS-CoV-2-specific T cells generated for adoptive immunotherapy are capable of recognizing multiple SARS-CoV-2 variants' has been provisionally accepted for publication in PLOS Pathogens.

Best regards,

Amy L Hartman, PhD

Associate Editor

PLOS Pathogens

Shin-Ru Shih

Section Editor

PLOS Pathogens

Kasturi Haldar

Editor-in-Chief

PLOS Pathogens

orcid.org/0000-0001-5065-158X

Michael Malim

Editor-in-Chief

PLOS Pathogens

orcid.org/0000-0002-7699-2064
---

## [Editor Report · Acceptance letter]

8 Feb 2022

Dear Dr. Smith,

We are delighted to inform you that your manuscript, "SARS-CoV-2-specific T cells generated for adoptive immunotherapy are capable of recognizing multiple SARS-CoV-2 variants," has been formally accepted for publication in PLOS Pathogens.

Best regards,

Kasturi Haldar

Editor-in-Chief

PLOS Pathogens

orcid.org/0000-0001-5065-158X

Michael Malim

Editor-in-Chief

PLOS Pathogens

orcid.org/0000-0002-7699-2064